

# Assessing climbers' pull-up capabilities by differentiating the parameters involved in power production

Marine Devise[1], Franck Quaine[2] and Laurent Vigouroux[1]

[1] ISM, Aix-Marseille University, CNRS, Marseille, France
[2] GIPSA, University Grenoble Alpes, CNRS, Saint Martin d'Hères, France

## ABSTRACT

This study explored the capabilities of sport climbers to pull up with arms. The methodology aimed at assessing (i) concentric capabilities of arm muscles, (ii) body coordination skills (iii) characteristics of energy storage and (iv) capabilities to resist fatigue. Twenty-eight climbers were tested and the force exerted was recorded during three pull-up exercises: jump tests (with or without coordination, or preceded by an eccentric phase), incrementally weighted pull-ups and maximum number of pull-ups. Force, velocity, muscle power and muscle work were analysed using ANOVA with post-hoc tests and principal component analysis. Correlations with climbing level were also studied. Overall, jump test results showed that body coordination and stretch-shortening cycle phenomena contributed significantly to performance but only the body coordination was related to the climber's grade level. Muscle work and maximum number of pull-ups are correlated with climbing level which showed that the capacity to resist fatigue is another crucial capability of climbers arms. The development of force capacities appeared crucial for performing whereas the velocity capabilities seemed to originate from the climber's own characteristics/style without correlating with climbing performance. Our study provides the basis for evaluating these parameters in order to help trainers in the diagnosis process and training follow-up.

## INTRODUCTION

Sport climbing requires multiple physiological capabilities, such as finger strength and endurance, upper and lower limb power, body flexibility or even body core strength (*Draper et al., 2021*). Among such capabilities, the capacity to pull up with the upper limbs is recognised as one of the main factors correlated with the climber's performance level (*Magiera et al., 2013*). Starting from a gripped hand hold, the upper limbs are used to displace the body from one position to the next one with the requirement that the contact between fingers and the hand hold is solid enough to support the amount of force needed to move the body (*Vigouroux et al., 2018*). Arm muscle action could be either isometric, concentric, eccentric or plyometric depending on the movements required by the particular route/boulder.

Corresponding author
Laurent Vigouroux,
laurent.vigouroux@univ-amu.fr

Assessing climbers' arm muscle capabilities has been a long-standing issue for trainers and researchers (*Stien, Saeterbakken & Andersen, 2022*). Basic tests consist of counting the number of successive pull-ups that the climber is able to achieve and/or measuring the amount of time that the climber is able to hang isometrically with a given flexion of the elbows, flexed to 90° for example (*Draper et al., 2021*; *Mermier et al., 2000*; *Stien et al., 2021*). Although these tests have been positively correlated with climbing grade level (*Draper et al., 2021*), the assessment is specific to a single level of power intensity for the pull-up test (*i.e.,* a given velocity with the body weight force level) or to a single angle tested for the isometric test. More recently, the maximum power developed by the arms has been proposed as a more appropriate variable for characterising the climber's arm capability (*Laffaye et al., 2014*). The power can be assessed by performing an explosive pull-up on either a power slap (*Draper et al., 2011*), a velocity transducer (*Muñoz López et al., 2017*) or an instrumented hangboard (*Vigouroux et al., 2018*). To go further in characterising the climber's capabilities, the relationship between force and velocity can be assessed (*Levernier, Samozino & Laffaye, 2020*; *Muñoz López et al., 2017*) by loading the climber incrementally until the maximum load that the climber is able to pull up is reached. When comparing with lower limb training, such relationships are essential to guide the athlete into suitable training that can focus on either velocity development or strength development according to the individual's profile (*Frost et al., 2016*). These assessments are thus crucial in assessing the concentric capabilities of the arm muscles.

Besides the movement of pull-up used as a basis for training and evaluation, this movement implies a complex combination of multi-joint movements (*Antinori et al., 1988*; *Ronai & Scibek, 2014*) and are not fully understood. Pull-ups can be influenced by the grip conditions such as grip size and hand orientation (*Lehman et al., 2004*; *Vigouroux et al., 2018*) and, more importantly, by the forms and the rhythm of completion (*La Chance & Hortobagyi, 1994*). *Vigouroux et al. (2022)* identified that pull-up performance is considerably influenced (+20%) by the stretch-shortening cycle when the specific pull-up is preceded by previous pull-ups. In particular, the elasticity storage characteristics, the transfer of movement quantity in between segments and the body coordination can have a strong influence on performance in addition to the concentric shortening capacities of arm muscles. This consideration leads to two main ideas. The first is that evaluations of arm capabilities should be carefully controlled (grip conditions, rhythms, leg and body coordination) to limit the variability in the results due to the various influences of all the phenomena involved in the movement. The second idea is that climbers probably do not only develop the arm muscle concentric capabilities to carry out the pull-ups and there is therefore an interest in differentiating each parameter (muscle concentric capabilities, stretch-shortening cycle characteristics, body coordination) to identify the strengths and weaknesses of a given climber.

The objective of this study was thus to assess the climbers' capabilities to pull up on holds. To this aim a series of tests was carried out to assess and characterise each pull-up parameter: arm muscle concentric capabilities, stretch-shortening cycle characteristics, body coordination capabilities, and arm muscle endurance. We hypothesised that such capabilities are correlated with the climbers' grade level. We also hypothesised that climbers
variously developed these characteristics and that such characteristics could be useful in characterising the different arm profile performance of climbers. Overall, we aimed to build an initial database of these parameters to help trainers and climbers identify their strengths and weaknesses.

## METHODS

### Participants

Twenty-eight male climbers (age: 28.4 ± 6.9 years; body weight: 66.2 ± 6.8 kg; height: 176.5 ± 5.4 cm) participated in the experiment. Inclusion criteria were to be aged from 18 to 45 years, to practice climbing at least twice a week for at least the past two years and to be from advanced to higher elite level of climbing according to *Draper et al. (2015)*. Participants practised both climbing disciplines (Lead and Bouldering) indoor and outdoor throughout the year. Each participant's red-point IRCRA level was determined for both bouldering and lead on the basis of self-reported best performances over the previous six months. For each participant, the highest level achieved in both disciplines was retained for the analysis (mean IRCRA red point level: 22.6 ± 2.5). 16 climbers presented a best IRCRA level for Lead while 11 had a best IRCRA level for bouldering, 1 climber had a similar IRCRA level between lead and bouldering. Exclusion criteria were to experiment hand, upper limb or spinal injuries in the last six months. All participants volunteered and signed an informed consent. The study was conducted with the formal approval of the CERSTAPS ethics committee (IRB00012476-2022-16-05-182).

All the tests were conducted during the same day. Participants were asked not to train or climb the day before the experiment and organize their training to be in the best possible shape for that experimental day. For all climbers, the experiment was scheduled away from competitions or heavy training period. An initial familiarisation session was carried out one week before testing and recording the data. This familiarisation session consisted in reproducing the exercises required in the experiment described below and used in order to avoid the discovery effect of both the exercises and the test equipment.

### Experimental protocol

Each participant followed a standardized warm-up. It consisted of a few easy climbs followed by incremental hang exercises performed on a 24-mm hold depth (ten 10s-hangs from 100N to full body weight) and five series of pull-ups (from one to four repetitions) executed on jugs. Participants ended the warm-up by testing one pull-up in each experimental condition detailed below. The experimental session consisted in executing successively three different types of exercise: jump tests, incremental weighted pull-ups, maximum number of pull-ups. The exercises were realized by participants in an order chosen to limit at best the effects of fatigue. Ten minutes of rest were respected between each of these three exercise types to avoid any effect of fatigue.

#### Jump tests

The first condition ("Strict Jump Test") consisted in an "explosive" two-armed pull-up performed as strongly as possible and as fast as possible from a completely hung static

position with arms extended and a prone grip. They put their hands always in the same place, slightly more widely spaced than their shoulders. At the top of the pull-up the participants were asked to jump and try to pursue the movement and not stop the movement once the chin is at the level of the hand holds as done in a power slap test. The participants were required to focus on the ascent phase while the down phase of the pull-up was executed comfortably to return to the ground without any performance objective. In this first condition, participants were asked to perform the jump in a "strict" manner –*i.e.*, only the arms were to be used to pull the body, without any use of leg and body coordination. In the second condition ("Normal Jump Test"), participants were asked to perform the same task again, but as they were accustomed to doing it, without any recommendations concerning any body coordination method *i.e.*, they were allowed to use legs and hips and a small swing as they were used to doing in their training. The third condition ("Countermovement Jump Test") differed from the others by the starting posture, which was static with the elbows flexed at 90°. Then the participants lowered themselves by "letting them fall" during an eccentric phase until their arms were extended (not fully but at least 120°) and then pulled up without delay. No constraining recommendations concerning body coordination were asked during this condition, participants being free to use legs and swing as they wished as in the Normal jump condition. Two trials were requested in each condition. The one with the highest mean power was retained for the analysis. For each trial, participants were motivated by experimenters to ensure maximum performance. At least 2 min rest separated each trial to avoid any effect of fatigue.

These three jumps were further used to evaluate the power and the peak force developed during the ascent phase of pull-ups including certain implied factors. The Strict Jump Test implies mostly concentric capabilities of the arm muscles, the Normal Jump Test implies the additional participation of body coordination and, finally, the Countermovement Jump Test adds the participation of the stretch shortening cycle phenomenon to the performance in comparison with the first two jump tests.

### Incrementally weighted pull-ups

The incrementally weighted pull-up exercise was conducted to assess the muscle concentric capabilities of participant's arms by building their force-velocity relationship. For this, participants were asked to perform several trials of one pull-up execution. The participants performed a "Strict jump" pull-up as strongly as possible and as fast as possible from a completely hung static position with arms extended. As for the jump tests, the hands were pronated and placed slightly more widely-spaced than their shoulders. The participants were required to focus on the ascent phase while the down phase of the pull-up was executed comfortably to return to the ground without any performance objective. The first pull-up was executed at body weight without additional load. Then, additional loads were added as a function of each participant's ability to achieve the newly weighted pull-ups. First increments were done with ten kilos, then in five-kilo increments and then in one-kilo increments until 1-RM (one-repetition maximum, *i.e.*, the maximum weighted pull-up) was reached. Two pull-ups per additional load were performed and the one with the highest

mean power was recorded. In total, 4 to 6 weighted pull-up conditions were executed. Each trial was separated by at least a 2-minute rest to avoid any effect of fatigue.

### Maximum number of pull-ups

The last exercise was a maximum repetition of pull-ups: participants were asked to do as many successive pull-ups as possible. They were required to perform each pull-up as fast as possible and as strongly as possible. The series stopped when pull-ups were not executed adequately *i.e.,* when executed with a rest in between each pull-up or with a too jerky velocity during the ascent phase, or when the participant was not able to place his chin over the hand holds.

## Materials

To measure the force exerted by the climbers during the tests, a SmartBoard (ScienceForClimbing, Peypin d'Aigues, France) was used as previously done in *Devise et al. (2022)* and *Vigouroux et al. (2018)*. SmartBoard is a hangboard fitted with force sensors (accuracy 0.8N, 0–4000N range, 50 Hz acquisition) measuring the vertical force applied on the holds. This tool makes it possible to collect the vertical force applied by the climber during the pull-ups with enough accuracy to calculate the physiological variables targeted in this study (vertical force, muscle power, muscle work). The associated app gives visual instructions to guide the participants during the tasks. The largest holds (jugs) were used for all exercises. As demonstrated by *Vigouroux et al. (2018)*, the use of jugs allows for arm performance at the same level as a gym bar without limiting finger strength. Force data were recorded during each test and then exported for post-acquisition analysis.

## Data analysis

Force data was low-pass filtered (Butterworth, fourth-order, cut-off frequency: 3 Hz). Data for each pull-up were re-sampled (100 points) for comparison purposes. Based on Newton's second law ($\sum \vec{F} = m.\vec{a}$), acceleration and velocity can be determined by using force data integration. Velocity and force are used to compute the muscle power as follow:

$$P(t) = F(t). \left( \Delta t. \frac{a(t+\Delta t) + a(t)}{2} \right) \tag{1}$$

where

$$a(t) = \frac{F(t) + BM.g}{BM} \tag{2}$$

where $F(t)$ is the force recorded by the force sensors, $a(t)$ is the acceleration of the system, $BM$ is the participant's body mass (kg), $g$ is the gravity acceleration ($-9.81$ m s$^{-2}$), $P(t)$ is the power. The mean power executed during the ascent phase of pull-ups was identified and expressed as a ratio to BM (W kg$^{-1}$).

For each trial, the mean force of the ascent phase was expressed (N kg$^{-1}$) as a function of the mean velocity (m s$^{-1}$) in order to compute the force-velocity relationship. A linear regression was used to evaluate the linearity of the relationship (r$^2$) for each subject and the slope of the force-velocity relationship. Theoretical maximum force (F0) and maximum velocity (V0) were estimated using the regression curves and correspond to the $y$- and $x$-intercepts of the curve with the force and velocity axes.

For the jump tests, the execution time for the ascent phase was determined. The maximum peak force and the peak power were identified in each condition and their timing as a percentage of the ascent phase duration was recorded. The mean force and the mean power were computed.

Finally, the number of pull-ups and the mechanical work (W) were evaluated during the maximum number of pull-ups test.

$$W(t) = \frac{\Delta t.(P(t + \Delta t) + P(t))}{2} \tag{3}$$

## Statistics

Descriptive statistics (Mean $\pm$ SD) were used to present the results of each variable. The statistical tests were processed with the use of the software STATISTICA (version 6; StatSoft, Inc, Tulsa, OK, USA). Each variable was correlated to IRCRA red point level using Pearson test correlations. One-way ANOVA with repeated measures and Newman-Keuls post-hoc tests were used to compare the three jump test conditions (Strict, Normal, Countermovement) for the mean force, maximum peak force, the timing of maximum peak force, the mean power, the maximum peak power, and the timing of maximum peak power, the time taken for the jump. The size effect ($\eta^2$) was computed and defined as small for $\eta^2 < 0.01$, medium for $0.01 < \eta^2 < 0.06$ and large for $\eta^2 > 0.14$ (*Olejnik & Algina, 2003*). Statistical significance was fixed at $p < 0.05$.

To identify the important variables contributing to the performance, a principal component analysis (PCA) was conducted using variables computed among the different tests (1-RM, F0, V0, Slope of force-velocity relationship, peak and mean force in each jump condition, peak and mean power in each jump condition, time of execution of each jump condition, maximum number of pull-ups, total energy). Number of principal components (PC) was determined according to the scree plot (>10%).

## RESULTS

### Normal, strict and countermovement jumps

Table 1 summarises the values obtained during the three types of jump. Figure 1A showed averaged force across subjects developed during the three tested jumps. ANOVA and post-hoc tests showed that the Countermovement jump was executed faster than Normal and Strict jumps (F(2,54) =66.9; $p < 0.001$; $\eta^2 = 0.34$) . The maximum force was significantly higher for the Countermovement jump than for the Normal jump which, in turn, was higher than that for the Strict jump (F(2,54) = 32.0; $p < 0.001$; $\eta^2 = 0.37$). The peak force was attained at different instants (F(2,54) = 73.4; $p <0.001$; $\eta^2 = 0.60$) between the Countermovement jump and the Strict and Normal conditions.

Figure 1B showed averaged power across subjects developed during the three jumps under test. The maximum peak power differed significantly (F(2,54) = 6.08; $p < 0.01$; $\eta^2 = 0.05$) according to the jump conditions and was reached at different instants of the ascent phase (F(2,54) =39.1; $p < 0.001$; $\eta^2 = 0.44$). The mean power developed during the jumps

**Table 1  Mean ± SD results of jump characteristics (duration, force values, power values) according to the jump form (Strict, Normal and Countermovement jumps).**

|  | Strict jump | Normal jump | Countermovement jump |
|---|---|---|---|
| Time of ascent phase (s) | $1.01 \pm 0.20$ | $0.97 \pm 0.19$ $^*r = -0.47$, $t = -2.7$, $p = 0.01$ | $0.72 \pm 0.13^{a,b}$ |
| Maximum Peak Force (N kg$^{-1}$) | $13.4 \pm 1.1$ | $15.1 \pm 2.6^a$ $^*r = 0.50$, $t = 2.9$, $p < 0.01$ | $16.6 \pm 1.3^{a,b}$ |
| Time of Peak Force (% of ascent phase) | $30.7 \pm 14.6$ | $31.0 \pm 13.0$ | $-2.2 \pm 10.9^{a,b}$ |
| Mean Force (N kg$^{-1}$) | $9.72 \pm 0.07$ | $9.73 \pm 0.13$ | $9.65 \pm 0.15$ $^*r = -0.45$, $t = -2.6$, $p = 0.015$ |
| Maximum Peak Power (W kg$^{-1}$) | $13.9 \pm 3.2$ | $16.1 \pm 4.9^a$ $^*r = 0.44$, $t = 2.5$, $p = 0.02$ | $15.0 \pm 3.9^a$ |
| Time of Peak Power (% of ascent phase) | $65.3 \pm 6.6$ | $51.9 \pm 15.5^a$ | $41.6 \pm 9.5^{a,b}$ |
| Mean Power (W kg$^{-1}$) | $6.8 \pm 1.0$ | $7.3 \pm 1.3^a$ $^*r = 0.42$, $t = 2.5$, $p = 0.02$ | $8.1 \pm 1.3^{a,b}$ |

**Notes.**
[a] Statistical difference with strict jump.
[b] Statistical difference with normal jump.
[*] Significant correlation with the climbing grade level.

was significantly different between the three conditions (F(2,54) =23.2; $p < 0.001$; $\eta^2 = 0.17$).

When correlating these variables with the IRCRA red point level of the participants, only the variables gathered during the Normal jump (maximum force, mean power, maximum peak power, time of jump) were significantly correlated (statistical results in Table 1). The mean force during the Countermovement jump was negatively correlated with the IRCRA level.

## Force-velocity relationship

Figure 2 presented the typical force-velocity relationships computed for participants. The 1-RM averaged $101.2 \pm 14.5$ kg. Mean individual $R^2$ averaged to $0.98 \pm 0.02$ and ranged from 0.95 to 1 showing a strong linear relationship between force and velocity for each individual among the incremental tests. The slope of curbs (N kg$^{-1}$/m s$^{-1}$) ranged from 6.4 to 21.2 with an average amounting to $11.1 \pm 3.1$. F0 averaged $17.8 \pm 2.9$ N kg$^{-1}$ and ranged from 11.9 to 26.4 N kg$^{-1}$. V0 averaged $1.65 \pm 0.19$ m s$^{-1}$ and ranged from 1.24 to 2.06 m s$^{-1}$. When correlating the variables from the force-velocity tests with the IRCRA red point level, V0 was not significantly correlated ($r = -0.26$, $t = -1.36$, $p = 0.18$ and $r = 0.03$,

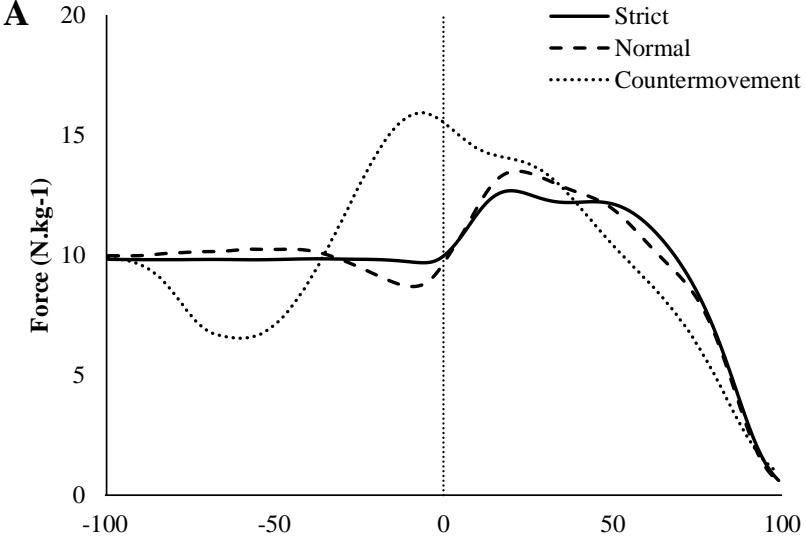

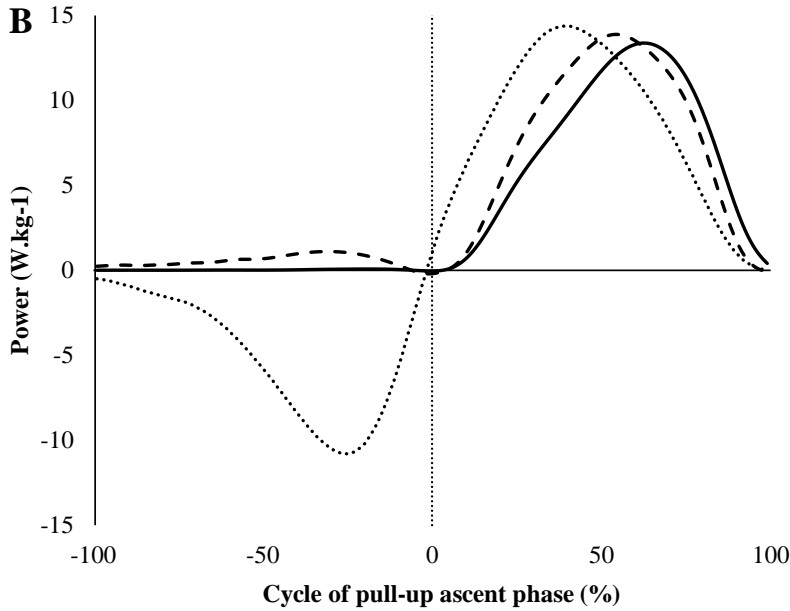

**Figure 1 Temporal evolution of the vertical force during jump exercises.** Temporal evolution (percentage of pull-up cycle) of the vertical force (A) and power (B) for the Strict (full line), Normal (dashed line) and Countermovement (dotted line) jumps. The pre-cycle (from −100 to 0%) is defined by the time before the beginning of the ascent phase.

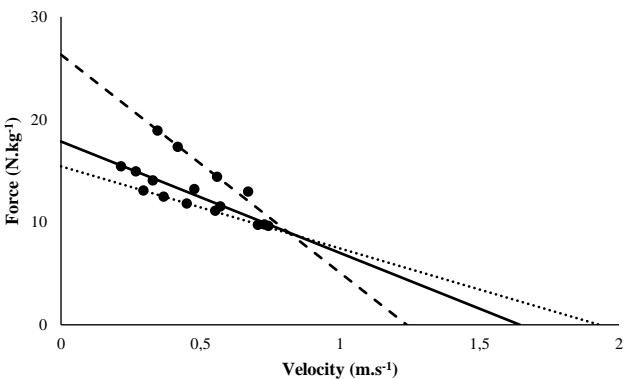

**Figure 2 Typical Force-Velocity linear regression from 3 climbers.** Typical force-velocity linear regression from 3 climbers. Climber 1 (dashed line) presented a high F0 but a low V0 whereas Climber 2 (dotted line) showed a low F0 but a high V0. Climber 3 (full line) had a mean F0 and a mean V0. F0: theoretical maximum force at null velocity; V0: theoretical maximum velocity at zero force.

$t = 0.14$, $p = 0.89$ respectively). The 1-RM ($r = 0.45$, $t = 2.58$, $p = .02$), the slope of the curve ($r = 0.39$, $t = 2.15$, $p = 0.04$) and F0 ($r = 0.43$, $t = 2.44$, $p = 0.02$) were significantly correlated to the IRCRA red point level.

### Maximum number of pull-ups

The maximum number of pull-up repetitions reached $22.5 \pm 7.7$ on average and ranged from 12 to 40. The mechanical work averaged $8{,}113 \pm 2{,}852$ J. These variables were significantly correlated with the IRCRA level ($r > 0.39$, $t > 2.2$, $p < 0.04$).

### PCA

According to the scree plot, two PC were retained for the PCA. The scree plot and projection of variables of the two PC are displayed in Figs. 3A and 3B, respectively. The first factor explained 44.4% of the variability. It polarised two types of variables: on the one hand, variables associated with time (time of jump execution, V0), mean force during Countermovement and Strict jumps; on the other hand, variables associated with peak force in the Countermovement and Strict jumps, F0, 1-RM, peak power, mean power, number of pull-ups, slope of force-velocity relationship, *etc*. The second axis explained 10.5% of the variability and especially discriminated variables associated to the timing of Normal jump (timing of peak force and timing of peak power) and the values of mean force and peak force during the Normal jump.

## DISCUSSION

The aim of this study was to assess the overall pull-up capabilities of climbers by conducting a series of three different exercises. The incremental pull-up tests were performed to analyse the concentric muscle power capabilities by distinguishing between muscle concentric force production and muscle velocity generation capabilities. The jump tests were performed to differentiate the capabilities of body segment coordination and the capabilities associated

**A**

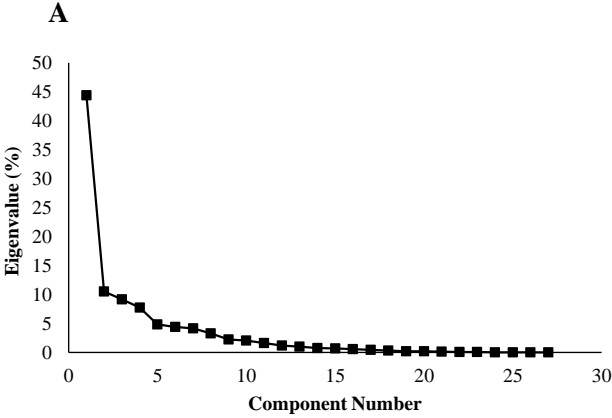

**B**

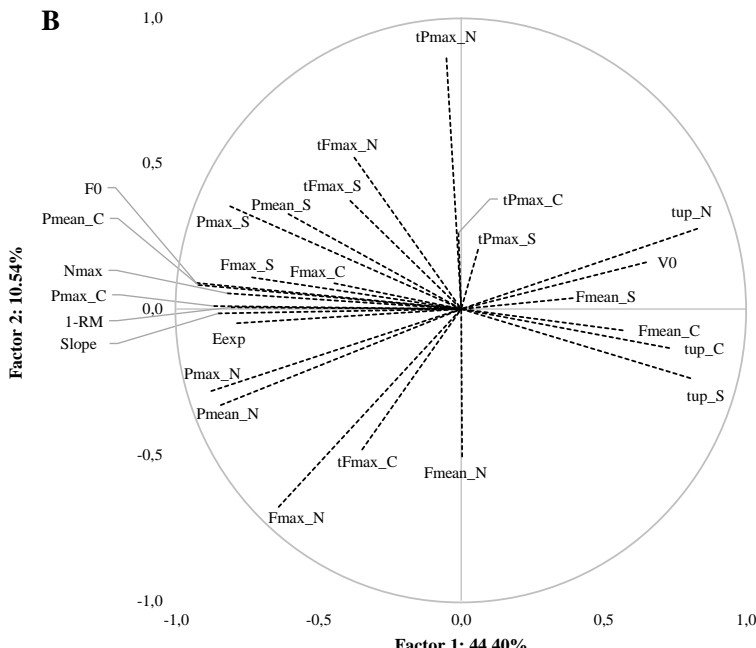

**Figure 3  Principal component analysis (PCA) results.** (A) Scree plot of eigenvalues after PCA. (B) PCA biplot of pull-up forms (S, Strict; N, Normal; C, Countermovement) and pull-up variables (Fmean, mean force; Fmax, peak force; tFmax, time of peak force; Pmean, mean power; Pmax, peak power; tPmax, time of peak power; tup, time of ascent phase of pull-up; Nmax, maximum number of pull-ups; Eexp, total energy expended; 1-RM, One-Repetition Maximum; Slope, slope of the force-velocity relationship; F0, theoretical maximum force; V0, theoretical maximum velocity). The proportion of variance captured is given as a percentage for both the first and second principal components (PC1 and PC2) which explained 55% of the variance.

to the stretch shortening cycle, both of which are additional contributors to the muscle power production. Finally, the maximum number of pull-ups was used to assess the ability to resist fatigue.

Regarding concentric muscle power generation, the incremental tests enabled developing the force-velocity relationship for each participant. Such relationships showed good linearity with high $R^2$ values. This confirmed previous research on force-velocity relationships on other musculoskeletal systems such as the lower (*Samozino et al., 2013*) and upper (*García-Ramos et al., 2016*) limbs and specifically those previously done during pull-ups (*Levernier, Samozino & Laffaye, 2020*; *Muñoz López et al., 2017*). Interestingly, for pull-up analysis, assessing these relationships using measurements of the vertical force applied at the hands, as we did in the current study, overcame the limitations associated with the measurement of acceleration or velocity at the centre of gravity identified by the authors mentioned above. Measuring the force exerted at the level of the hands is indeed a less noisy signal and is not influenced by the swinging movement of the body during the pull-ups as is the case for sensors placed at the level of the body's centre of gravity (*Levernier, Samozino & Laffaye, 2020*).

The current results of force-velocity relationships showed that the slope of the curb is highly dependent on the climber's characteristics since we observed values ranging from single to triple. This is in agreement with *Levernier, Samozino & Laffaye (2020)* who showed that elite boulderers presented relationship slopes significantly different from lead climbers ($-10.5$ and $-14.2$ for elite lead and elite boulder climbers respectively). This slope could thus be easily used by trainers to determine the climbers' need for either force or velocity gain, according to the positioning compared with the values reported in the current study ($-11.1 \pm 3.2$ N kg$^{-1}$/m s$^{-1}$). Deeper in the analysis, it is notable that only the variables associated with force (1-RM, F0 and the Slope) are correlated with the climbing grade level. This means that the force is a crucial capability to develop for improving the climbing level. On the contrary, the velocity parameter (V0) is not correlated with the climbing grade level which suggests that this parameter is associated with a characteristic not related to the level but that that could distinguish the climbers such as the climbing strategy or the time management during the climb. A potential parameter that can discriminate these strategies is the climbing fluency identified by computing the jerk of the 3D movement of the hips during climbing (*Seifert et al., 2014*). In their study, these authors showed that the route characteristics (type of holds) and the number of repetitions can affect the climber fluency. Further research might be conducted in order to investigate the potential correlation between climbers' fluency and the physiological arm characteristics identified with the method of the current study. Overall, differentiating the various capabilities of muscle power generation (force *vs.* velocity) appears thus crucial in assessing the climber's arm muscle properties and needs.

In addition to the arm concentric muscle characteristics, the results of jump tests showed that other phenomena also contribute to the performance since the comparison of jump performances (peak force, mean power, peak power, time of execution) are deeply influenced by the different types of jumps. Especially, the velocity of the jump is higher in Normal (with body coordination) and Countermovement (with prior eccentric phase allowing using a stretch-shortening cycle) jumps than in the Strict jump condition. The mean power developed during the ascent phase improved by 7.3% between Strict and Normal jumps and then by 11.0% between the Normal and Countermovement jumps.
The peak force and the peak power were attained earlier with the Countermovement jump whereas those in the Strict jump condition were attained later than those in the Normal jump. *Vigouroux et al. (2022)* observed similar performances when comparing separated pull-ups (with a pause between each pull-up) to pull-ups executed continuously (with no pause between the downward movement and the subsequent upward movement).

These improvements between the different jump types are in line with results observed when comparing different jumps with the lower limbs (*Van Hooren & Zolotarjova, 2017*, between countermovement and squat jumps). This suggests that similar phenomena identified in the lower limbs are at work in the upper limbs, such as the use of "opposite" limb swing and stretch-shortening cycle including storage and utilisation of elastic energy, the residual force enhancement, the stretch reflex, the reduction of muscle slack and the build-up of muscle stimulation (*Van Hooren & Zolotarjova, 2017*). Given that participants were hanging with arms extended before executing the Strict and Normal jumps and with arms flexed before the Countermovement one, the phenomena of stretch-shortening cycle should contribute considerably to the observed differences between the Countermovement and the other jumps. Nevertheless, since arm muscle tendons present anatomical differences in length and properties for energy storage and utilisation of elastic energy in comparison with the lower limbs (*e.g.*, Achilles tendon), these aspects probably contribute to a lesser extent in the upper limbs. In addition, the difference observed between Strict and Normal jumps suggests that movement of other limbs (leg swing, trunk movement) and the body coordination are crucial for enhancing performance, as has been observed in the lower limbs with the use of the arm swings during a leg squat jump (*Hara et al., 2006*). For enhancing pull-up capabilities, there is thus a strong interest in developing those skills (body coordination and stretch-shortening cycle).

Interestingly, the variables associated with the Normal jump are correlated with the climbing grade level, whereas those associated with the Strict and Countermovement jumps are not (or negatively). These apparently contradictory results provide complementary information. They showed that the skills solicited for the Strict jump (muscle contractile properties only) and those solicited for the Countermovement jump (stretch-shortening capabilities) are not related to the climbing level: climbers with a low grade level can perform similarly to climbers with a high grade level. On the other hand, performance in the Normal jump is related to the climbing grade level, which means that using body coordination in conjunction with arm muscle contraction is an important skill for climbers and a good indicator of their climbing grade level. A possible logic is that the climbers do not manage to achieve their optimal muscle performance when they must pull with the arms alone (as in the Strict jump) or with a previous eccentric phase (as in the Countermovement jump). Climbers probably need to mobilise their body coordination in synergy to provide their full arm capabilities. Overall, it can be conducted that the body coordination skills are an important part of the climber's performance. Such parameters should therefore be investigated, using the Jump tests presented in the current study, in addition to the concentric muscle arm capabilities, which are themselves characterised by the force-velocity relationship.

Concerning the maximum pull-up exercise until exhaustion, the maximum number of pull-ups and the energy expenditure were correlated with the climbing grade level as previously demonstrated by *Draper et al. (2011)*. The current results reinforced those previous findings and showed that the capacity of arms to resist fatigue is a crucial parameter for performance. Nevertheless, unlike these authors, we tested a maximum number of repetitions without rhythm instructions: the pull-ups were linked without recommendation on the speed to adopt. *Draper et al. (2011)* recommended performing pull-ups with a constrained rhythm (2 s for the ascent phase, 2 s for the descent phase). Given the importance of coordination and the stretch-shortening cycle that are demonstrated in this study, it seemed more relevant not to give rhythm instructions in order to assess the effect of the combined factors determining pull-up performance on the participant's level of fatigue.

When combining all parameters together, the PCA showed two important facts that should be taken into account when considering the arm capabilities for climbing. First of all, 44.4% of the variability between climbers can be explained by the distinction between velocity parameters of the pull-ups and muscle force parameters (including those of fatigue resistance). Nevertheless, it is important to consider that the velocity parameters are not correlated with the climber grade level. It thus seems possible to have a solid grade level without high velocity qualities but the force parameters and the resistance to fatigue are unavoidable factors correlated with the grade level. Consequently, it seems that some climbers differentiate themselves from others by the velocity parameters without having any incidence on the final climbing performance. As discussed above, the velocity parameters may thus be associated with climber's strategy and performance management. This idea is consistent with the second factor of the PCA which differentiates timing parameters and amplitudes of peak power and peak force during the Normal jump. For practical application this means, firstly, that training focusing on force development is essential when aiming to improve the overall climbing level. Secondly, training focused on velocity parameters appears to be more essential when aiming to adapt or modify the climber's strategy profile for a specific boulder/route objective. To go further in this analysis, an interesting perspective might be to evaluate the "overall efficiency" during pull-ups as an integrative parameter to understand the performance as it is done for locomotion (*Peyré-Tartaruga & Coertjens, 2018*).

In our study we chose to test climbers practising both disciplines (lead and bouldering) both indoor/outdoor in order to test climbers in the perspective of a combined climbing sport (bouldering and lead) as is the case for the Olympic Games in Paris in 2024. To deal with this diversity of practice, we evaluated the participants' IRCRA level by retaining the best IRCRA level observed in bouldering/lead. 16 climbers presented the best grade level in lead, while 11 climbers are better in bouldering (1 climber had a similar level in both disciplines). This could be seen as a limitation as *Levernier, Samozino & Laffaye (2020)* identified differences in force-velocity relationships between elite lead climbers and elite boulderers. To ensure that this point did not impact our results, we performed a final additional statistical analysis, which consisted of comparing the two parts of our participants using t-tests. No statistical difference was found in any of our variables

($p > 0.05$). This confirms that for advanced to higher elite climbers who practise both disciplines simultaneously, the grade level of practice and individual characteristics are more important than the preferred form of practising the sport.

Some limitations should be considered for this study. First, completing all of the tests by itself was a great achievement for the climbers. That is why we only tested climbers with a minimal advanced level of climbing performance (minimal of 7a red point). This choice was made to avoid participants not able to complete the tests because they are not sufficiently trained in pull-ups. A second reason was to avoid having participants well trained in the practice of other sports (such as cross-fit) but with a non-matching level of climbing performance. Hence, our results only address the climbing level that we tested and further studies should be conducted for lower climbing levels. Similarly, additional studies should be performed to investigate the pull-up capabilities of female climbers, as they are known to have different physiological characteristics from men for a similar climbing performance level (*Mermier et al., 2000*). A second limitation concerns the estimation of the velocity, which was determined by integrating the force signal. Such integrating processes are known to amplify the low frequency noise. It could thus be possible that velocity is over- or under-estimated. Although this necessarily leads to less error than estimating force by deriving velocity, the best experimental solution would be to combine both force and velocity measurements at the same time.

Overall, this study showed that the power production with arms in climbers originates from multiple phenomena. Even when testing the climbers on the seemingly simple pull-up movement, different parameters emerged as crucial, and the analysis of the muscle power amount alone appears insufficient to fully assess the climber's arm characteristics. In particular, body coordination during jumps contributes significantly to the performance and is related to the climber's grade level. The capacity to resist fatigue is also confirmed as important. Finally, when analysing the concentric muscle capabilities, the development of force capabilities appeared crucial for performing while the velocity parameters are not related to the climbing performance but could originate in the climbing strategy profile. Our study has proposed a method to collect these parameters to help trainers in the diagnosis process of the climber. For example, by performing the three proposed exercises (Jump tests, incrementally weighted pull-ups, maximum number of pull-ups), a trainer/climber can identify his own arm strengths and weaknesses and his own positioning in the ACP profile. This diagnosis could be then used to elaborate a training strategy and to post-assess the effects of a training period. From a fundamental point of view, an interesting perspective of this study is to use the proposed method to evaluate the benefits of different types of training.

### Funding
This research has received funding from ANR PPR STHP 2020 (project PerfAnalytics, ANR 20-STHP-0003). The funders had no role in study design, data collection and analysis, decision to publish, or preparation of the manuscript.

## Grant Disclosures

The following grant information was disclosed by the authors:
ANR PPR STHP 2020: PerfAnalytics, ANR 20-STHP-0003.

## Competing Interests

Laurent Vigouroux reports his involvement in the development of SmartBoard by SmartBoard Climbing, and his current position as scientific adviser to SmartBoard Climbing. SmartBoard Climbing was not involved in any aspect of this research. The remaining authors declare that the research was conducted in the absence of any commercial or financial relationships that could be a potential conflict of interest.

## Author Contributions

- Marine Devise conceived and designed the experiments, performed the experiments, analyzed the data, prepared figures and/or tables, and approved the final draft.
- Franck Quaine conceived and designed the experiments, prepared figures and/or tables, authored or reviewed drafts of the article, and approved the final draft.
- Laurent Vigouroux conceived and designed the experiments, performed the experiments, authored or reviewed drafts of the article, determined the evaluated factors for analysing the pull-ups, and approved the final draft.

## Human Ethics

The following information was supplied relating to ethical approvals (i.e., approving body and any reference numbers):

The study was conducted with the formal approval of the CERSTAPS ethics committee (IRB00012476-2022-16-05-182).

## Data Availability

Raw data are available in a Supplementary File.

## Supplemental Information

Supplemental information for this article can be found online at http://dx.doi.org/10.7717/peerj.15886#supplemental-information.

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
