# Peer review of "Assessing climbers’ pull-up capabilities by differentiating the parameters involved in power production"

_PeerJ, doi:10.7717/peerj.15886_

## Round 0.1 · original submission · Major Revisions

The authors aimed to analyze the capabilities of sport climbers to pull-up with arms in three different exercises.

The paper is well written and structured, with the main objective clear.
However, the reviewers and I have found several points to adjust.

ln 28-30 - this sentence is tricky. be objective in results (mainly) that don't present relation.

ln 73-33 - Consider including in this paragraph (and into the discussion) the problem of elasticity taking into account muscle and overall movement efficiency. I suggest reading this article (PMCID: PMC6297284) especially this section:
…Both contractions have different efficiency values. W+ denotes the work performed to raise and accelerate the center of body mass utilizing concentric contractions, whereas W- denotes the work performed to decelerate and reduce the height of the center of body mass employing eccentric contractions. Another important aspect is that W- causes more repercussions in the calculation of Wext than those in the calculation of Wint. In these cases, the efficiency of the W-, which influences Wext directly, is considered. Thus, to control the effect of Wel from the “apparent efficiency” calculation, subtraction needs to be performed only from Wext (Minetti et al., 1994a). Furthermore, Wel is only a component of W- (Minetti et al., 1994a).
Efficiency can be expressed in various other ways, such as the term effmec. Although initially employed in the investigation of the isolated muscle (Hill, 1913), studies related to effmec were later extended to full-body research (Hill, 1922)….

·

Basic reporting

According to the authors, the present study aimed to assess the climbers´ capabilities to pull up on holds by conducting a series of three different exercises. The main findings of the authors were that body coordination and capacity to resist fatigue were related to climbing grade.

Language is clear, and the context is adequately clarified. However, I have a few comments.

Experimental design

1. Line 87. The purpose of the study needs a revision. It is established as part of the objective to determine elasticity storage capabilities of the climbers, however, they are not mentioned again in the rest of the study.
2. Participants section need to be completed “Twenty-eight male climbers (age: 28.4±6.9 years; body weight: 66.2±6.8kg; height: 176.5±5.4cm) participated in the experiment. They practised climbing at least twice a week for at least the past two years and were from advanced to higher elite level of climbing (mean IRCRA red point level: 22.6±2.5 according to Draper et al., 2015)”. kind of climbing did the participants practise (outdoor/indoor)? How many participants practice boulder and lead? You provided details in the discussion as a limitation but not in the description of the sample. Did the sample's climbing ability correspond to their level in boulder / lead and indoor/outdoor climbing? Please provide participants´ climbing ability (grade level) in boulder and lead climbing, whether they practiced both disciplines and if they practiced one of them.
3. What are the implications for the results that all the participants have practiced both climbing disciplines (main and bouldering) with half the group favoring one or the other?
4. How did the selection of climbers for the study group proceed?
5. Please provide the inclusion and exclusion criteria in the research.
6. Please specify how the sample size was calculated for statistical analysis.
7. Why was the SmartBoard (Science For Climbing, Peypin díAigues, France) chosen to measure pull-ups on holds? Are there studies on the reliability of this device? please add references or explain your choice.
8. The force was measured with force sensors on the handboard, however, how was the velocity parameters measured? were accelerometers used?

Validity of the findings

9. Lines 342-344. “Interestingly, the variables associated with the Normal jump are correlated with the climbing grade level while those of the Strict and Plyometric jumps are not (or are negatively).”. How do the authors explain these contrary results?
10. Were all tests performed on the same day? Considering that maximum number of pull-ups and incrementally weighted pull-ups as maximum force test does this not affect the results? Clarify the number of visits of participants for the completion of the study and the procedures in each visit.
11. Has it been taken into account how finger strength affects the results?
12. Lines 302-304- The sample size is too small and heterogeneous in relation to the type of climbing (bouldering / lead) to use the values of this study as reference values. In addition, the study did not take into account how the weight, body mass and age of the participants affect the results.
13. Lines 308- The authors should explain what mean climbers´ style or climbers´s type. Clarifying the concepts will give the readers a more comprehensive view of the explained.
14. Line 314. “Especially, the velocity of the jump is higher in Normal ..” should be rewritten as “Especially, the velocity of the jump was higher in Normal”
15. Lines 368-369. Velocity parameters may thus be associated with a climbing style or a climbers´ type. The conclusion of the study focusses on that velocity parameters may be associated with climbing style or type, but such conclusion cannot be withdrawn from the statistical results. The type of analyses and study design does not allow such statement.
16. Lines 376-377. “In our study we chose to test climbers practising both disciplines (lead and bouldering) with half of the participants favouring one or the other. This was done in order to test climbers in the perspective of a combined climbing sport (bouldering and lead) as is the case for the Olympic Games in Paris in 2024. This could be seen as a limitation since Levernier et al. (2020) identified differences in Force-Velocity relationships between elite lead climbers and elite boulderers.”. Why weren't both groups of climbers compared? Please provide details on the level of the participants in each of the disciplines.
17. Lines 381-384- Nevertheless, in our study, due to the combined practice of our participants, no such statistical differences were observed (p>0.05) indicating that for intermediate climbers, the grade level of practice and the individual characteristics are more important than the preferred form of practicing the sport. In this paragraph you indicate that there are no differences because they are intermediate climbers, while in the methods section you detail that the average level of the participants corresponds to an advanced to higher elite level of climbing. Please clarify this inconsistency.
18. Abstract: Review according to changes throughout the manuscript. Contribution to the field: Needs revision: "Our study provides the basis for some reference values for those parameters to help trainers in the diagnosis process and training follow-up for climbers".
19. Please provide implications for practice of the results obtained.

Reviewer 2 ·

Basic reporting

The presented research aimed to evaluate the capabilities of sport climbers to pull-up with arms.
The manuscript is well structured, with the main purpose so clear.
All factors are described clearly. However, I have several comments.

Participants:
1. "They (sport climbers) practised climbing at least twice a week for at least the past two years and were from advanced to higher elite level of climbing (mean IRCRA repoint level 22.6±2.5) - All participants practised both climbing disciplines (Lead and Bouldering) with half of the group privileging one or the other".
- How was the IRCRA level calculated for climbers practicing both climbing subdisciplines - based on the result in bouldering or lead climbing?
- Was the best RP score taken from the whole career or the last 3 or 6 months?
- Was the result considered during rock climbing or on an artificial wall?
2. "They had no hand or upper limb injuries in the past six months and were asked not to train the day before the experiment".
- Were spinal injuries (cervical, lumbar) considered during subjects selection? Spinal injuries can affect the velocity and movement pattern during pull-ups.
- Intense training two days before the measurements may have also influenced the study results.
- Please provide whether climbers were during the competition season or whether training cycles were taken into account (e.g., whether someone was currently training or had a rest or was in the general preparation phase)
- Please specify whether the study group also included climbers competing in bouldering competitions - which may require different training and type of climbing (more dynamic and coordination elements - which may also be related to the obtained results).

Experimental design

The work is original and considers innovative research tools (SmartBoard). The study is well-designed and reproducible. The only limitation regarding the study group was given above.

Validity of the findings

The results are essential for athletes and coaches regarding the specificity of pull-ups. They also constitute the basis for further research in this area. Consider dividing participants into boulderers and lead climbers. Moreover, it is interesting how these results would look if speed climbers were included.

Additional comments

In general, the work is interesting and can contribute to the sport climbing literature.
However, I think my comments about the study group should be addressed in the paper before publication. I hope my suggestions will help improve this work.

---

## Round 0.2 · accepted · Accept

The previous Academic Editor is no longer available and so I am taking over handling this submission in my capacity as Section Editor.

Thank you for addressing the Reviewer's comments and making the necessary changes to your manuscript. I am pleased to recommend your amended manuscript for publication. Thank you for supporting PeerJ, and we look forward to future manuscript submissions from you and your co-authors.

·

Basic reporting

I consider that the authors should deeply review the english writing of the article, since I have found throughout the text several incorrect grammatical expressions. I think that verbs that indicate certainty are used throughout the text when verbs that indicate the possibility of it being that way should be used "May, might and could".

Line 28. “number of pull-ups are correlated with climbing level which showed that the capacity to..” should be re-written as “number of pull-ups were correlated with climbing level which could showed that the capacity to”....

Lines 348-350. Please revise the ingles redaction “is not correlated with the climbing grade level which suggests that this parameter“ should be rewritten as “was not correlated with the climbing grade level which might suggests that this parameter “

Line 349 there is one more “that”

Line 394 “Normal jump are correlated with” should be re-written as “Normal jump were correlated with”

Line 396 “Countermovement jumps are not (or negatively)” should be re-written as“Countermovement jumps were not (or negatively)”

Lines 397-400 “They showed that the skills solicited for…” Who showed? This is probably wrong wording and it is missing references.

Line 399 “are not related to the climbing level”, I consider it should be revised and re-writte as “were not related to the climbing level”

Line 401 “performance in the Normal jump is related” should be re-written as “performance in the Normal jump was related”

Line 408 “Overall, it can be conducted change I consider that this is your conclusion or recommendation so it should be rewritten as “Overall, it could be conducted….”

Line 490. What ACP profile means? These acronyms are the first time they appear, please clarify their meaning.

Experimental design

The study is well-designed and replicable and limitations have been detailed in the manuscript.

Validity of the findings

The manuscript and results will be useful for climbers and coaches.

Additional comments

In general, I feel the work show interesting insights that will contribute to improve the sport climbing knoledge. However, according to my comments, I think that the english wording should be reviewed and improved before its publication.

Reviewer 2 ·

Basic reporting

Thank you for taking into account all my comments.
Congratulations on the interesting paper.
In its current form, the work meets the criteria for publication.

Experimental design

The work is original and considers innovative research tools (SmartBoard). The study is well-designed and reproducible.

Validity of the findings

The results are essential for athletes and coaches regarding the specificity of pull-ups. They also constitute the basis for further research in this area.

Additional comments

I just wanted to let you know that there are no additional comments.
Thank you.